# Hybrid Fiber Materials according to the Manufacturing Technology Methods and IOT Materials: A Systematic Review

**DOI:** 10.3390/ma16041351

**Published:** 2023-02-05

**Authors:** Hye Ree Han

**Affiliations:** Department of Beauty Art Care, Graduate School of Dongguk University, Seoul 04620, Republic of Korea; luckyherry@hanmail.net

**Keywords:** hybrid fiber material, electrical conductivity, shape memory, sputtering, electrospinning

## Abstract

With the development of convergence technology, the Internet of Things (IoT), and artificial intelligence (AI), there has been increasing interest in the materials industry. In recent years, numerous studies have attempted to identify and explore multi-functional cutting-edge hybrid materials. In this paper, the international literature on the materials used in hybrid fibers and manufacturing technologies were investigated and their future utilization in the industry is predicted. Furthermore, a systematic review is also conducted. This includes sputtering, electrospun nanofibers, 3D (three-dimensional) printing, shape memory, and conductive materials. Sputtering technology is an eco-friendly, intelligent material that does not use water and can be applied as an advantageous military stealth material and electromagnetic blocking material, etc. Electrospinning can be applied to breathable fabrics, toxic chemical resistance, fibrous drug delivery systems, and nanoliposomes, etc. 3D printing can be used in various fields, such as core-sheath fibers and artificial organs, etc. Conductive materials include metal nanowires, polypyrrole, polyaniline, and CNT (Carbon Nano Tube), and can be used in actuators and light-emitting devices. When shape-memory materials deform into a temporary shape, they can return to their original shape in response to external stimuli. This study attempted to examine in-depth hybrid fiber materials and manufacturing technologies.

## 1. Introduction

The emergence of the next-generation Fourth Industrial Revolution is, at present, promoting research on artificial intelligence (AI), the Internet of Things, information and communications technology (ICT), intelligent fibers, nanowires, and smart materials. Consequently, smart wear is becoming an item that will dominate the fiber material industry in the future. Hybrid fiber composites can have various applications, such as in healthcare, defense, fashion and entertainment, sportswear, purpose clothing, and transportation, as well as integration with advanced technology [1,2,3,4,5,6,7,8]. To date, research on cutting-edge hybrid fiber materials is being conducted [4,5,9,10,11,12,13,14,15,16,17,18,19,20,21,22,23,24,25,26,27,28,29,30,31,32].

Materials used in smart textiles include shape-memory materials, metal fibers, conductive inks, nanoparticle optical fibers, organic semiconductors, chromic materials, and inheritance-conductive polymers. McCann introduced phase-change materials, thermochromic materials, shape-memory alloys, quantum tunneling composites for switching devices, light-emitting polymers, photovoltaics and solar cells, photoluminescence, plasma technology, microencapsulation for therapy delivery, global positioning, wireless communications, radio-frequency-identification (RFID) tags and microelectronic mechanical systems (MEMSs), and exoskeletons [33,34]. S. Lam Po Tang et al. presented smart-clothing technologies in their research. They suggested that shape-memory materials/polymers, phase-change materials, chromatic materials (thermochromic and photochromic dyes), stimuli-responsive hydrogels and membranes, and smart wearable electronics (conductive materials, flexible sensors, wireless technology, and alternative power sources) could be used as smart technology in the textile industry [35]. Langston et al. described the IOT(Internet of Things) and smart products that enable data to be transferred to the cloud [36]. Li et al. explained that conductive fabrics form an electronic conductive network, becoming one of the basic elements of wearable electronic fabrics that interconnect various electronic functional units, such as fiber electrodes and power sources [37]. Kamalakann et al. argued that the rapid increase in IOT knowledge enables the attainment of additional information for potential applications. In particular, the information on interoperability methods is obtained by collectively considering various smart objects across the internet. It also shows that resource-based IOT access technology is useful in various information environments to support information on cloud-computing platforms [6]. Kamalakann et al. showed that the research at present shows additional applications in industries, such as IOT and healthcare [6]. Senem Kursxun Bahadrr commented on smart clothing designed for medical applications, which can typically monitor heart rate, respiration rate, ECG (electrocardiogram), and temperature. It was also added that sensors should be placed in the correct location of smart clothing to accurately collect information about users’ health [38]. The potential benefits of smart medical textiles include the integration of functionality into the textile interface, flexible materials that conform to the body, wearable materials suitable for clothing or bedding, and familiar interfaces providing more comfortable products and versatility in design and materials. Additionally, these textiles allow for the continual monitoring of mobile patients or post-operative recuperation, as well as for premature or chronically ill babies and elderly patients. Furthermore, it can aid the reduction in invasive procedures, decrease power utilization for communication, and provide cost-effective solutions appropriate for disposable usage [33,34]. Innovative fusion functions can, additionally, be added to existing electrically conductive materials using shape-memory polymers, electrospinning, sputtering, and 3D printers. 

In this study, the practical state of hybrid fiber materials and manufacturing technologies are investigated. In addition, foreign data on manufacturing technology and intelligent IOT materials are analyzed. Furthermore, this study examines various cutting-edge materials used for high-technology hybrid fiber materials and considers methods of convergence, thus providing a foundation for future hybrid fiber material technology.

With the development of ICT (information and communication technology) and material technology, the wearable device industry is rapidly expanding. Various studies, such as those concerning sensors and monitoring, are being conducted [39,40,41,42,43,44]. 

The major international standards established in TC124 related to wearable technology are listed in Table 1 [45]. TC124 (wearable electronic devices and technologies) has set the goal and scope for establishing international standards for wearable electronic devices and technologies, including accessory-type and electronic fiber materials and devices.

The fabrication of smart fabrics can be divided into coating and lamination processes. Coating methods include dip, knife or blade, air knife, metering rod, transfer, roll, paste dot, and powder [46]. Laminating methods include flame, wet adhesive, hot melt, dry heat, and ultrasonic. Flame lamination is a process in which a prepared thin thermoplastic foam sheet is passed over an open flame to generate a thin layer of a molten polymer. Polyurethane foam (PUF) is the most frequently used foam. Wet adhesives used in the laminating process are either water- or solvent-based. They are applied to the substrate surface in liquid form using conventional coating methods, such as gravure roll coating, spraying, roll coating, and knife coating. Then, the adhesive-coated web is bonded with other substrates under pressure and dried or cured in an oven [46]. An additional potential method involves intermixing a metal powder (conductive powder) into a yarn and adding the electrically conductive thread into the yarn when weaving or knitting the warp (or weft). It can be 3D printed and coated onto a fabric. A 3D-printed material is considered durable once the remaining components (such as an appropriate binder) have been applied. 

The latest materials used in smart textiles include metal fibers, conductive inks, electrospun nanowebs, shape-memory materials, nanoparticle coatings, optical fibers, organic semiconductors, chromic materials, and conductive polymers. Manufacturing techniques used in smart textiles include sputtering, 3D printing, and electrospinning.

In this study, a systematic review was conducted. Figure 1 presents the research identification process conducted through the database presented in this study. Literature research data were collected using Web of Science, SCOPUS, Google Scholar, and Riss. Keywords used in researching the literature included IOT material, electrical, conductivity, conductive material, nanofiber, sputtering, shape memory, and 3D printing.

Articles, editorials, and journals not related to the subjects were excluded. The database lists of the included research were further screened for additional eligible publications. Additionally, the selected 61 papers were divided into three compositions and reviewed in depth.

Recently, demand for multifunctional materials is increasing due to the advent of high-tech and the advancement of consumer needs. However, there are very few data that have been systematically classified and organized by collecting extensive data on hybrid fiber materials and production technologies. Therefore, this study attempted to examine in-depth hybrid fiber materials and manufacturing technologies.

## 2. Manufacturing Techniques of IOT Hybrid Fiber Materials

### 2.1. Sputtering

Systematization of sputtering journals with bibliographic source are listed in Table 2. Sputtering technology thinly coats metal onto the fiber and has an eco-friendly advantage as it does not generate wastewater. In addition, fibers that have introduced sputtering can be used as military stealth materials, smart wear using electrically conductive materials, and artificial intelligence materials [47,48,49,50]. 

Additionally, there is a study in which metal nanograins, such as aluminum, copper, and nickel, were formed on the fabric through sputtering treatment [51]. The metal layer of the magnetron sputtering fabric rapidly emits the body temperature into the open air, concealing the body in infrared thermal-imaging cameras [52]. However, the effect of stealth technology depends on the sputtering processing time; therefore, the sputtering process must be performed for an appropriate period [51,52]. In addition, a flexible and wearable electrically conductive pressure sensor was developed using SnCl_4_ treatment and Ag sputtering on nylon. The manufactured pressure sensor was observed to be highly reproducible and repeatable for 9500 repeated mechanical loads, with a low capacitance loss rate of 0.0534. Fabric-based flexible and comfortable sensors can be integrated into fabric garments using thermal pressure. Conductive nylon fabric in the twill structure, which showed a high conductivity rate of 0.268 Ω/cm (specific resistance), was prepared by magnetron sputtering with silver films. The flexible pressure sensor exhibited a high sensitivity value of 0.035 kPa^−1^ [53].

Sputtering technology is advantageous as it is environmentally friendly, has a simple manufacturing process, and produces no wastewater compared to other forms of coating technology. In addition, it has stealth technology, electrical conductivity, and electromagnetic wave blocking in which the thickness of the layer can be easily adjusted according to process changes. Therefore, as it is so versatile, it can be used as a state-of-the-art hybrid fiber in a variety of fields. 

Applicable examples of using sputtering technology include semiconductors, automotive parts, heavy industries, stealth clothing materials, electromagnetic wave blocking materials, electrically conductive materials, and sensors.

### 2.2. Electrospinning

Electrospinning is becoming increasingly popular in the field of the production of nanofibers [54]. Systematization of sputtering journals with bibliographic source are listed in Table 3. Electrospinning products can be used for protective materials, structurally colored fibers, self-cleaning materials, adsorbents, electromagnetic shielding, agriculture, low-temperature proton-exchange membrane fuel cells, solid oxide fuel cells, hydrogen storage, supercapacitors, lithium-ion battery materials, dye-sensitized solar cell applications, biosensors and biocatalysis, wastewater treatment, and air pollution control [55,56,57]. 

The thickness of the electrospun nanoweb was varied to manufacture membranes with different pore diameters. There are three main types of electrospinning devices. The first is a “high-voltage power”, which is usually 50 kV, and the second is “spinneret”, where the nozzle radiating speed is an important factor in determining fiber thickness. The third is the ink collector. The distance between the tip and collector determines the degree of elongation and the fiber thickness. Several studies have been conducted to regulate electro-radiation conditions for various variables. 

Bokova et al. addressed fiber electrical rotation technology for nonwoven fabric production in various applications. In particular, they studied the conditions for forming nano- and microfibers in collagen hydrolysate and dibutyrylchitine solutions, as well as polymer complexes based on polyacrylic acid, polyvinyl alcohol, and polyethylene oxide. Comparative analyses of electrical rotations, electrical capillary tubes, and electrical nano spiders were performed. The results show promise not only for garment and shoe production, but also for the application of nonwoven fabrics in pharmaceutical hygiene practices [58]. Kang et al. successfully fabricated alumina nanofibers with a diameter of 100–800 nm using an electrospinning AlCl_3_/PVP solution. In addition, they systematically investigated the structural and microstructural properties of alumina fibers using analytical tools, such as SEM, TG-DTA, FTIR, XPS, and XRD. As a result of this investigation, high-quality non-crystalline Al_2_O_3_, g-Al_2_O_3_, and a-Al_2_O_3_ nanofibers were obtained by calcining AlCl_3_/PVP hybrid fibers for 5 h at 450, 900, and 1100 °C, respectively [54]. Laszkiewicz et al. discussed the possibility of nanofiber formation of different solvent systems with electrostatic fields in cellulose solutions. The samples obtained through electrospinning were observed via SEM, and the spinning dope was obtained using N-methylmorpholine, N-oxide, and viscose methods. Based on studies conducted with phosphoric acid mixtures not commonly used in the industry, Laszkiewicz et al. argued that the cellulose solution of NMMO(N-Methylmorpholine N-oxide) could be demonstrated to be a good system for the stable formation of nanofibers using static electricity [3].

He et al. suggested thermoelectric nanofiber yarns using CNT/PEDOT: PSS with electrospinning technology, which showed high stretchability (~350) and sealability. In addition, self-powered strain sensors consisting of threads show corresponding thermal voltage changes according to strain, which can be used to optimize the shooting rate of basketball players. These unique features have shown that thermoelectric nanofiber seals have a wide range of prospects in smart wearables such as wearable generators, respiratory monitoring, and motor optimization [59].

He et al. proposed a state-of-the-art manufacturing strategy that combines electrospinning technology and spraying technology to manufacture (2022) carbon nanotube (CNT)/polyvinylpyrrolidone (PVP)/polyurethane (PU) composite thermoelectric fabric. As a result, a self-powered sensor for detecting joint motion has been successfully proposed. In addition, the electrical conductivity and Sebeck coefficient did not change after 1000 bending times [60].

In addition, recently, studies conducted by electrospinning using hybrid materials mixed with the two have been actively conducted.

Amjadi et al. studied carbon-loaded zein/pullulan hybrid electrospun nanofibers for food and medical use. Zein/pullulan hybrid nanofiber was produced in three mixing ratios: 90:10, 80:20, and 70:30 using the electrospinning method. The tensile strength and water contact angle values of the zein/pullulan/carvone nanofiber were 7.09 ± 0.85 MPa and 96.6 ± 0.7°, respectively. In addition, the incorporation of carbon into NF provided DPPH(α-diphenyl-β-picrylhydrazyl) elimination activity (9.6 ± 1.2%) and inhibitory activity against S. Aureus (39.95 ± 3.81 mm) and E. coli (12.2 ± 1.4 mm) bacteria. The biocompatibility of the developed NP has been approved by in vitro cytotoxicity analysis. In conclusion, the developed electrospinning NFs have significant potential in several applications, such as food packaging and wound dressing [61].

Chiu et al. used a double nozzle electrospinning process side by side to manufacture flexible hybrid electronics (FHE) materials with excellent elasticity. Very stable electrical conductivity was also provided to the membrane electrode produced by using silver nanoparticles (AgNP) and carbon-based nanomaterials having different structures. The AgNP/carbon nanomaterial was coated on a binary polymer nanofiber composed of polyurethane (PU) and polyvinylidene difluoride (PVDF) on a nanofiber membrane. FHE nanofiber electrodes were finally incorporated into garments designed to accurately measure human body detection signals (e.g., ECG and EMG signals). AgNP/graphene oxide (GO) nanofiber electrodes showed a continuous phase with a stable material microstructure after 5000 iterations of a 50% tension–tension fatigue test. The waveform patterns obtained from the proposed AgNP/GO nanofiber electrodes were compared with conventional ECG and EMG electrodes. Nanofiber web electrodes treated with organic/inorganic mixed dispersants and verified through electrical and fatigue characteristics tests are suitable for long-term ECG and EMG monitoring and have excellent potential in wearable smart sensors [62].

As such, electrospinning can be used in various fields to produce nano-diameter fibers and pores. Electrospinning conditions change nanoweb characteristics with the variables of voltage, tip-to-collector distance, collector rpm, temperature, humidity, and pH. Therefore, if personalized hybrid fibers are required, electrospinning technology can meet consumers’ needs.

Applicable examples of using electrospinning technology include filtration, protective clothing with high moisture vapor transport, toxic chemical resistance, breathability fabric, tissue engineering applications, wound dressings, fibrous drug delivery systems, implants, transdermal patches, and nanoliposomes, etc.

### 2.3. Three-Dimensional Printing 

Three-dimensional printing is a process that uses additive materials, and the starting products are manufactured by stacking the layers individually. Thus, the CAD model was physically reproduced by individually stacking materials upwards from the bottom. Cross-sectional data is required to create objects because the product is manufactured using the program. Systematization of sputtering journals with bibliographic source are listed in Table 4.

Therefore, the process uses STL files, which are 2D files that can be 3D printed by creating 3D models using single sections. In addition, 3D printers reduce the cost and time required for prototyping as they simplify manufacturing processes, thereby reducing labor and assembly costs. Instead of products, digital drawings can be distributed and printed anywhere [63,64].

Three-dimensional printing methods are largely solid-, liquid-, and powder-based. Solid-based models include fused deposition modeling (FDM), fused filament fabrication (FFF), and LOM (laminated-object manufacturing).

Fused deposition modeling (FDM), which is the most frequently used 3D printer, is mainly used by PLA(Poly Lactic Acid) and TPU(Thermoplastic Polyurethanes). The FDM-type filament is formed by stacking the material that is melted in the heated extruder and flows out of the nozzle onto a plate. As this method does not use a laser, it has the advantages of being a simple mechanism, having high durability and strength properties, and efficient manufacturing cost and time.

Liquid-based models include SLA (stereolithography apparatus), DLP (digital light processing), Polyjet (photo polymer jetting), and MJP (multi-jet printing).

The digital light processing (DLP) method uses a liquid photocurable resin. Liquid materials are placed in a tank where light can be transmitted, and parts are selectively cured by projecting cross-sectional images of the sculpting object onto the material, using the DLP engine. The DLP method has the advantage of producing low noise levels and sophisticated products; however, if the production size increases, the resolution decreases.

Powder-based models include selective laser sintering (SLS), DMLS, 3DP, DED, PBF, SLM, PBP, and EBM.

In the selective laser sintering (SLS) method, a solid-particle powder material is used. The laser light source is selectively irradiated over the material, which is flattened by a leveling roller, causing it to be partially melted and bonded. Nylon is the primary material used, as it has strong rigidity and high-temperature properties. However, it requires a preheating or cooling process, which is inconsistent in dimensional stability and allows for limited color properties [65,66,67,68].

Three-dimensional printers are being used in a variety of fields at present, such as core-sheath fibers, vascular diseases, artificial organs, and intelligent textiles. In addition, previous studies have proposed a 3D-printing model that can be used for valve, vascular, and structural heart diseases [64]. In another study, core-sheath-coated fiber patterns were manufactured using a 3D printer equipped with coaxial spinnerets. Smart patterns were manufactured using ‘CNT’ as a core and ‘silk fibroin’ as a dielectric sheath. As a result, biomechanical energy can be obtained from human movement with a high-power-density value of 18 mW/m^2^ [32].

Applicable examples of using three-dimensional printing technology include artificial organs, skulls, fetal figures, electronic parts, cars, buildings, shoes, accessories, and dresses, etc.

## 3. IOT Hybrid Fiber Materials

With the advent of artificial intelligence and IT technology, this paper will examine materials that can be used in IOT hybrid fibers that respond to external stimuli or have electrical conductivity.

### 3.1. Conductive Materials

With the recent increase in the number of cases of combining electronic devices with clothing, the research surrounding this subject has improved [7,13,69,70,71,72,73,74,75,76,77]. Systematization of sputtering journals with bibliographic source are listed in Table 5.

Productive materials for wearable electronics are available on the market, including conductive yarn, fiber, ribbon, paint, and tape. Metals have several advantages, such as low electrical resistance, and many studies have employed these properties. Metal materials (for example, silver, copper, and stainless steel) are coated on yarns or fabrics to provide electrical conductivity. Recently, with the rapid development of IT devices and increased interest in smart textiles and the Internet of Things (IoT), the necessity for electro-conducting textiles has increased. In this context, emphasis is placed on flexible materials that decrease electric resistance through the application of metal nano-substances. These textiles present less functional damage against deformations, such as bending and tension. In advanced studies, metal nanowires have been widely used in the material science sector within coating technology. This is due to their excellent conductivity and simplicity in processing, which is beneficial for flexible displays and the electronics industry. In addition, studies trialing metal nano-substances’ applications to textiles to examine their antibiotic actions and UV-blocking features have also been conducted.

Silver nanowires (AgNWs) are percolating network nanostructures that have been widely used as flexible and folding conductors. Shih Pin et al. combined AgNW network connections, using simple chemical reactions, to increase the mechanical strength of AgNW thin films under extreme stretching conditions. The soldered nanostructures strengthened the conductivity of the network. Additionally, it showed no significant change in electrical conductivity during the roll-pressing process or elongation, up to an elongation strain of 120%. Their potential applicability in flexible electronic devices has also been demonstrated in the research [78]. Polyaniline (PANI) and polypyrrole (Ppy) are also widely used as electrically conductive materials, and are mainly used in sensors, actuators, batteries, and fuel cells in smart wearable devices and are produced by melt-spinning. PANI has favorable characteristics, such as low cost, safety, processability, and ease of synthesis [79]. PPy is an insulator, and its oxidative derivative is a good electrical conductor. The 2000 Nobel Prize in Chemistry was awarded for the study of conductive polymers, including polypyrrole. The PPy research example is as follows.

Carlo Emilio Standoli et al. used ICT to study PEGASO system technology. The system also proved that the reliability of evaluating changes in youth lifestyles and the effectiveness of sensor network systems can be assessed [80]. Heo et al. presented a wearable textile antenna embedded in a wireless power transfer system. A planar spiral coil was fabricated using conductive threads on a cotton substrate and connected to a rectifier circuit assembled on a flexible polyethylene terephthalate film to form a receiver that could be bent using magnetic resonance. At a resonance frequency of 6.78 MHz, the proposed system was able to achieve 5.51 dB transmission efficiency and 12.75 mW power transmission at a distance of 15 cm [81]. Őscar Belmonte-Fernández et al. located users based on Wi-Fi intensity signals received from nearby wireless access points. They also presented an indoor positioning system for wearable devices based on Wi-Fi fingerprints [82]. Song et al. synthesized conducting cellulose–acrylamidoxime compounds and studied their electrical conductivity properties. In particular, the electrical conductivity of Cu_x_S–cell–AA (copolymer Cu_x_S–cellulose–acrylamidoxime, Cu_x_S: activated sulfur atom reaction after the introduction of copper 1 ion) was greater than that of Cu_x_S–cell–AN (copolymer Cu_x_S–cellulose–acrylonitrile) complexes. DSC analysis showed that cell–AN has increased thermal stability compared to that of cell–AA, and Cu_x_S–cell–AA had a Tm value approximately 18 °C lower than that of Cu_x_s–cell–AN, which is thought to have promoted pyrolysis due to its high thermal conductivity [71]. Seo et al. studied the pore characteristics of activated carbon aerogels (ACAs) by conducting KOH and CO_2_ activation of carbon aerogels. Additionally, they considered the electrochemical characteristics by applying them as electrodes in electric double-layer capacitors (EDLCs). CA (carbon aerogel) activated with KOH (potassium hydroxide) had a high specific surface area and high EDLC electrode specifications, but it had developed more mid-air than micropores, a relatively considerable reduction in specific capacity at a high current, and low electrical conductivity due to most of the crystallinity loss occurring in the activation process. In contrast, CO_2_-enabled CA had a specific surface area half of that of KOH-enabled CA, but it had a specific capacity of approximately 80% [83]. Yang et al. studied the improvement of thermal conductivity by adding high-k (TCA) and aluminum oxide (Al_2_O_3_) to the resin layer of artificial leather at a consistent rate. HTCAL’s k value was 48.3 % higher than that of regular leather [84].

Meanwhile, numerous cases using electrically conductive substances as sensors have been reported. Standoli et al. conducted a study on smart wearable sensor systems to combat lifestyle diseases, such as obesity, in teenagers. In this work, they studied standardized experimental protocols for testing signal reliability in smart clothing prototypes. Additionally, they considered methodologies through joint design activities and approaches to address user requirements and preferences, as well as technical specifications. The UCD approach has been proven to be effective and reliable in presenting solutions that meet users’ needs and preferences. It also explained that these wearable systems were effective in assessing lifestyle and changes in selected teenagers (such as diet and physical activity) [80]. Bahadir et al. explained that, to accurately collect information about users’ health, it was essential to place sensors in the correct location in smart clothing. In this study, the appropriate sensor position for smart clothing was determined using acceleration measurements to detect the internal organs and muscle disorders of the wearer. Accelerometers were installed in different clothing locations to determine breathing difficulties, heart disease, and mental disorders, and to measure respiratory rates, heart rates, and muscle tremors. Experiments have shown that sensors located on the shoulder area of smart clothing structures are best positioned for obtaining information on respiration rate, heart rate, and muscle tremors [38]. Grym et al. studied the practicality of smart wristbands that can be continuously monitored during pregnancy and for one month following childbirth. It would be useful to use a maternity care wristband for the continuous monitoring, tracking, and transmission of biometric signals to individuals. However, they explained the necessity to carefully evaluate the factors of device design and comfort [12].

However, when EM-blocking (electromagnetic blocking) aprons for pregnant women are available on the market, the effects of electromagnetic waves on mothers and fetuses (such as side effects) due to smart clothing should also be considered. Romare et al. provided a medical professional’s perspective on the use of smart glasses during intensive care in a qualitative research method. Smart glasses equipped with customized ICU (intensive care unit) software have increased monitoring accessibility, indicating that monitoring and patient safety can be improved during intensive care. However, the process of implementing smart glasses should be prudent and help facilitate patient-centered treatment and patient safety when ICU staff feel confident and comfortable with the technology. In addition, this study explained that smart glasses were proposed to complement existing monitoring methods and routines, and they cannot replace human activity during intensive care [11]. Li et al. explained that the combination of fiber and electronics has become a promising area for future fiber development. However, the increased use of wearable electronic fabrics in the market has been delayed. As a result of focusing on textile functionality, products have poor aesthetics at present. Consumers then regard the product as “technology” rather than “clothing”, because it has overlooked aesthetics [37]. Belmonte et al. presented a Wi-Fi fingerprint-recognition positioning system based on machine learning algorithms and wearable devices. In smart watches, users can estimate their current location using a robust, stable, and easy-to-use wearable device. In this study, they empirically tested the location tracking and battery life of users during their waking hours. The sensor explained that it can be used for specific purposes, such as fall detection [82]. Zhang et al. proposed a new method to recognize human emotions (such as neutral, happiness, and anger) using smart bracelets with built-in accelerometers. They also confirmed that it is possible to accurately recognize a person’s emotions (such as neutral, happy, and angry) using wearable devices. When classified into two categories, the accuracy values were 91.3 % (neutral dialogue), 88.5 % (neutral versus happiness), and 88.5 % (happiness dialogue). These results have been argued to be useful in improving human–computer interaction performances [15]. Yang et al. studied high-thermal-conductivity artificial leather (HTCAL). The material and structure of the base layer were modified to incorporate smart electronic fiber materials (conductive yarn) into the production of traditional artificial leather. Consequently, the thermal conductivity of HTCAL improved by 19.6 %, and the temperature difference between the skin and the inside of the leather increased when worn directly, making it comfortable and cool to wear in the summer [85]. Bedeloglu et al. investigated the development of photovoltaic PP (polypropylene) fibers using two types of polymer-based photoactive materials. This study permitted highly conductive PED OT: PSS (Poly(3,4-ethylenedioxythiophne): poly(styrenesulfonate)) solutions and translucent metal layers to be successfully used as positive and negative organic solar cells for flexible devices, respectively [86].

Li et al. studied conductive stitches modeled by distributed resistance networks consisting of length-related and contact resistors. Analytical equations were derived to model a complex resistive network of conductive knit-stitching and conductive knitwear techniques. These were built from conductive yarn under natural relaxation or regular extension conditions on a major or minor axis of fabric. In this study, it was shown that the equivalent resistance of the conductive threads of knitwear could be modeled by equations that significantly simplified the existing model [87].

Carbon nanotubes are also of interest due to their robust characteristics. CNTs have electrical conductivity properties similar to copper, thermal conductivity approximately equal to diamonds, and strength 100 times greater than steel. Carbon nanotubes have hexagonal shapes consisting of six carbons connected to form a tube, which is several tens of nanometers in diameter. Many studies have been conducted on the application of semiconductors, flat-plate displays, and batteries. Wang et al. introduced ultrasonic and flexible total-textile air-volume sensors based on fabric-grown carbon nanotubes (CNTs) in the field by imitating hair on spiders for use in electronic skins and smart textiles [88]. Liang et al. reported that most carbon nanotubes have low dispersibility properties and suspected cytotoxicity in solvents, making them difficult to use in ink. It was explained that sericin-CNT (SSCNT) ink can be used as fiber, paper, plastic, electrocardiogram, and respiration sensors using various techniques [89]. Carbon nanotubes (CNTs) include single-wall nanotubes (SWNTs) and multi-wall nanotubes (MWNTs), which are combined with other polymer materials to compensate for polymer shortcomings [90,91,92]. Additionally, carbon nanotubes are being studied at present for the manufacturing of microporous foam, electrical conductivity, thermal properties, gas permeability, and relative electrodes for dye-sensitized solar cells [93,94,95,96]. Jung et al. added SSFs and MWNTs to polypropylene, manufactured composites using a biaxial screw extruder, and studied their mechanical properties and surface resistance. As a result of measuring the surface resistance of the complex by varying the MWNT content, a percolation threshold of 1–2 wt% occurred. It was observed that adding 1 wt% SSF to 1 wt% MWNT without percolation resulted in a rapid decrease in the resistance [97].

Chen et al. explained that CNTs are also potential candidates for understanding discrete artificial synapses. LTP (long-term potentiation) or LTD (long-term depression) was induced by the increased conductivity as a result of the dehydrogenation of CNTs. LTP or LTD was not obtained when the gap between two consecutive spikes was wider than 20 ms. This was because the hydrogen ions in PEG (poly(ethylene glycol)) returned to the equilibrium position within that interval. The CNT synaptic device consumed only 7.5 p J/spike, which is compatible with the widespread integration of artificial neural networks [98].

Graphene is lightweight, has high strength, and has excellent electrical conductivity. Therefore, many studies have recently been conducted on this material. The research conducted on MWCNTs and SWCNTs has also increased. The lengths are commonly below 5 µm, while the diameters are approximately 2 nm for SWCNTs and between 2 and 15 nm for MWCNTs [72,98,99,100,101]. Shemshadi improved the thermal properties of polyacrylamide (PAAm) membranes using graphene oxide (GO) and reduced graphene oxide (rGO) while utilizing them as static modifiers and phase-change material microcapsules (mPCMs). It was reported that the presence of reduced graphene oxide improved the heat-absorption performance of mPCMs. These hydrogel membranes improved protective clothing comfort [102].

The demand for electrically conductive products is rapidly increasing, as is the research. Therefore, it is important to select a material suitable for the environment and improve washing durability. Although electrically conductive materials have various advantages, human friendliness, such as electromagnetic wave blocking, should also be considered.

### 3.2. Shape Memory

Shape-memory materials can recall their original shape. When the material deforms into a temporary shape, it can return to its original shape in response to external stimuli. The shape change is activated most often by changing the surrounding temperature, but with certain materials, stress, magnetic field, electric field, pH value, UV light, and even water can be the triggering stimuli [103,104,105,106]. Systematization of sputtering journals with bibliographic source are listed in Table 6.

Shape-memory polymers (SMP) present more limited transformation ranges than shape-memory alloys, but have the advantages of low density, mass, and price, while being easily manufactured and dyed. Intelligent textiles with optimal environmental responses were developed from electrospun nanowebs with controllable pore structures, using spinning conditions. The use of shape-memory polyurethanes allowed the shape of the material to be retained and recovered through heating. Examples of shape-memory-material research are as follows:

As an SMP material used in an FDM (fused deposition modeling)-type 3D printer, there is a case in which ‘Partially neutralized poly(ethylene-co-r-methacrylic acid) ionomers (Surlyn by Dupont)’ were extruded into a filament and used as a model thermoplastic resin-shaped memory material [107]. Razzaq et al. used aluminum nitride (AIN) particles to increase the thermal conductivity of commercial SMPUs. At a 40 wt% loading value of AlN, the thermal conductivity increases from 0.12 to 0.44 W(m)^−1^. However, the inclusion of AlN significantly reduces the shape-recovery rate from 97 % to 70 % [105,108]. Matsumoto et al. observed the memory capability of an electric-sponge nonwoven fabric manufactured from two determinable multiblock copolymers, an x-pentadecalactone hard segment (PDL) and e-caprolactone switching segment (PCL), named PDLCL. As a result, the PDLCL multiblock copolymers exhibited good shape-memory properties with a strain recovery rate of 89–95% and a strain fixation rate of 82–83% after two cycles with a small strain applied at directly related temperatures [109]. Michalak et al. studied a multilayer fiber fabric prototype using NiTi and Cu as shape-memory devices. The fabric prototype developed was composed of three layers of nonwoven fabric created from a mixture of flax and steel fibers, two layers containing spirals, and composed of NiTi or reference copper (Cu) wires. At approximately 35 °C, changes in the temperature-dependent curve for a heating time of prototypes using NiTi elements could be observed, and the heating rate began to decrease. The width of the interlayer, including the air and NiTi elements, increased by approximately 2.5 mm during the heating process. This phenomenon was caused by the expansion of the NiTi spiral and not by prototypes composed of inert reference Cu elements. During the final minute of heating, the exterior surface temperature of the prototype using the NiTi element was 2 to 3 °C lower than that of the prototype using the Cu element. A theoretical model of the system was developed, and a satisfactory agreement was achieved between the experimental and theoretical results [110]. Michalak et al. studied prototypes of smart fiber materials with shape-memory elements that provide variable thermal insulation according to heat emission absorption. The shape-memory components are constructed in a spiral form of bidirectional action on NiTi unidirectional wires. Active spirals expand at temperatures below the characteristic internal state-transition temperature and contract when the temperature exceeds the transient temperature of approximately 45° [111]. Meng et al. knitted fabric to produce shape-memory fibers. The prepared samples presented much higher mechanical-strength properties than the corresponding shape-memory films, owing to the molecular orientation influenced by the rotation process. Studies have shown that shape-memory fabrics are not cytotoxic, hemolytic, sensitive, or irritable. These shape-memory fibers can be applied to artificial tendons, orthodontics, scaffold materials, round dressing, artificial corneas, hernia repair, and artificial bone joints [104].

As previously described, various studies have been conducted on shape memory owing to its characteristic memory and recovery properties. Therefore, if this performance is considered, it can be used as a state-of-the-art smart textile that can respond to varying external stimuli.

## 4. Summary and Future Prospective

In this study, the status of hybrid fiber materials and manufacturing technologies were analyzed, according to the material domain, and the application in the manufacturing of hybrid fiber products. An especially systematic review was also conducted. This includes sputtering, electrospun nanofibers, 3D (three-dimensional) printing, shape memory, and conductive materials. This study has shown that hybrid fiber material applications, such as sensors, vascular, e-textiles, devices, valves, and intelligent textiles were investigated. The characteristics and requirements of these high-performance materials could pave the way for future hybrid fiber materials.

Current status.

Sputtering technology is an eco-friendly, intelligent material that does not use water. In the case of sputtering, it can be applied as an EM shield, conductive fabric, and stealth material. Electrospinning technology using high voltage can be applied to breathable nanowebs and artificial blood vessels, etc. by manufacturing hybrid nanofibers. Three-dimensional printing includes FDM, DLP, SLS, and LOM methods. It is projected that 3D printing will be used in various fields, such as core-sheath fibers.

Conductive materials can be used in sensors, EM blocking, and wearable computers, etc. and well known conductive materials include silver nanowires (AgNWs), polypyrrole (Ppy), PANI (polyaniline), and carbon nanotubes (CNTs), etc. When the shape-memory material deforms into a temporary shape, it can return to its original shape by changes, such as pH and temperature, etc. Shape-memory materials can be used for intelligent textiles, and surgical sutures, etc.

The main obstacles to be applied.

In the case of electrospinning technology, ‘limitations of mass production, durability, and manufacturing speed ‘ have been raised as an obstacle.

In addition, in the case of 3D printing, the problem of toxicity is raised during the manufacturing process.

In the case of electrically conductive materials, when electricity is applied to fabrics, they should be worn only for a short period or with a small area in consideration of the harmfulness of electromagnetic waves directly touching the human body. Electrically conductive smart wear materials could have possible side effects on the human body. Additionally, further research should be conducted on the adequate use of space and time, and the prevention of electrocution during lightning storms.

In the case of shape memory materials, durability, etc., should be considered when repeatedly used.

Future perspective.

As the demand for multifunctional materials increases, it is expected that more multifunctional hybrid fibers will emerge in the future, which are more human-friendly, faster in production, and have good strength and washing durability. In particular, conductive fabric that combines technology and arts, fiber with improved medical function, hybrid fiber with increased human affinity, drug delivery system fiber with increased absorption rate, breathable fabric, artificial blood vessels, artificial organs, electrical conductivity fiber with more sensor sensitivity and biodegradable SMP hybrid fibers are expected to increase.

Having made these brief points, it is time to bring this paper to a close and to end. Hybrid fiber materials are expected to significantly contribute to several areas of research. In this study, a foundation for artificial intelligence was suggested by analyzing the characteristics of hybrid fibers and their manufacturing technology methods, recent applications, demand, and related high-performance materials.

## Figures and Tables

**Figure 1 materials-16-01351-f001:**
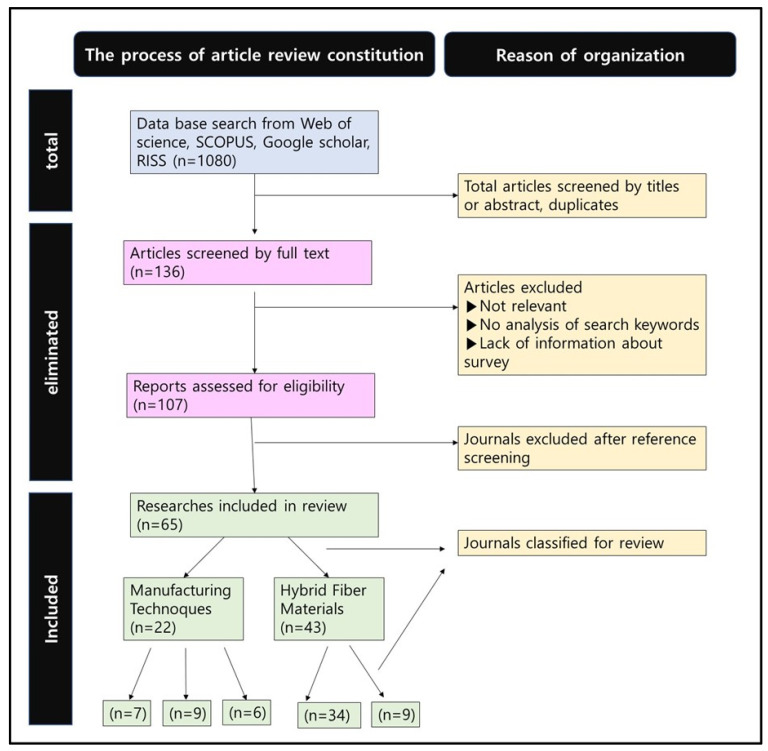
The identification process of research via databases.

**Table 1 materials-16-01351-t001:** Status of major international standards being established in TC124 related to wearable technology [45].

Standard Number (Include Status)	Standard Name	Proposed Year
IEC 63203-101-1:2021	Wearable electronic devices and technologies-Part 101-1: Terminology	2021
IEC 63203-204-1:2021	Wearable electronic devices and technologies-Part 204-1: Electronic textile-Test method for assessing washing durability of leisurewear and sportswear e-textile systems	2021
IEC 63203-401-1 ED1	Wearable electronic devices and technologies-Part 401-1: Devices and Systems–Functional elements-Evaluation method of the stretchable resistive strain sensor	2019
IEC 63203-402-1:2022	Wearable electronic devices and technologies-Part 402-1: Performance measurement of fitness wearables-Test methods of glove-type motion sensors for measuring finger movements	2022
BS EN IEC 63203-406-1:2022	Wearable electronic devices and technologies. Test method for measuring surface temperature of wrist-worn wearable electronic devices while in contact with human skin (IEC 63203-406-1:2021).	2022
IEC TR 63203-250-1:2021	Wearable electronic devices and technologies-Part 250-1: Electronic textile-Snap fastener connectors between e-textiles and detachable electronic devices	2021

**Table 2 materials-16-01351-t002:** Systematization/enumeration of sputtering papers with bibliographic source.

Authors	Title	Bibliographic Source
Su C-I et al. [47]	Performance of viscose rayon based activated carbon fabric modified by sputtering silver and continuous plasma treatment	Text. Res. J
Depla D et al. [48]	Smart textiles: An explorative study of the use of magnetron sputter deposition.	Text. Res. J
He S et al. [49]	Surface functionalization of Ag/polypyrrole-coated cotton fabric by in situ polymerization and magnetron sputtering	Text. Res. J
Yuan X et al. [50]	Electrical and optical properties of polyester fabric coated with Ag/TiO_2_ composite films by magnetron sputtering	Text. Res. J
Han H R et al. [51]	A study on thermal and physical properties of nylon fabric treated by metal sputtering (Al, Cu, Ni)	Text. Res. J
Han H R [52]	Characteristics of infrared blocking, stealth and color difference of aluminum sputtered fabrics	J. Korean Soc. Cloth. Text
Wu et al. [53]	A facile method to prepare a wearable pressure sensor based on fabric electrodes for human motion monitoring	Text. Res. J

**Table 3 materials-16-01351-t003:** Systematization/enumeration of electrospinning papers with bibliographic source.

Authors	Title	Bibliographic Source
Kang W et al. [54]	2011 A new method for preparing alumina nanofibers by electrospinning technology	Text. Res. J.
Cavaliere S et al. [55]	Electrospinning for Advanced Energy and Environmental Applications	CRC Press
Ding B et al. [56]	Electrospun Nanofibers for Energy and Environmental Applications	Springer
Dávila S M et al. [57]	Challenges and advantages of electrospun nanofibers in agriculture: A review	Mater. Res. Express
Bokova E S et al. [58]	Obtaining new biopolymer materials by electrospinning	Fibres Text. East. Eur
He X et al. [59]	Continuous manufacture of stretchable and integratable thermoelectric nanofiber yarn for human body energy harvesting and self-powered motion detection	Chemical Engineering Journal
He X et al. [60]	Highly stretchable, durable, and breathable thermoelectric fabrics for human body energy harvesting and sensing	Carbon energy
Amjadi S et al. [61]	Development and characterization of the carvone-loaded zein/pullulan hybrid electrospun nanofibers for food and medical applications	Industrial Crops and Products
Chiu CW et al. [62]	2022 Flexible Hybrid Electronics Nanofiber Electrodes with Excellent Stretchability and Highly Stable Electrical Conductivity for Smart Clothing	ACS Appl. Mater. Interfaces

**Table 4 materials-16-01351-t004:** Systematization/enumeration of three-dimensional printing papers with bibliographic source.

Authors	Title	Bibliographic Source
Zaharin H A et al. [63]	Additive Manufacturing Technology for Biomedical Components: A review	IOP Conf. Ser.: Mater. Sci. Eng.
Yi R et al. [64]	Delta DLP 3-D printing of large models	IEEE Trans. Autom. Sci. Eng.
Jung J-t et al. [65]	3D printer operators.	EBS broadcasting textbook Book Publishing Agency Gun Gi Won
Korea I et al. [66]	3D printer operations technician	Yeamoonsa.
Encyclopedia [67]	Selective laser sintering	Wikipedia the free encyclopedia
Giannopoulos A A et al. [68]	Applications of 3D printing in cardiovascular diseases	Nat. Rev. Cardiol

**Table 5 materials-16-01351-t005:** Systematization/enumeration of conductive materials papers with bibliographic source.

Authors	Title	Bibliographic Source
Quinn B [69]	Textile Futures: Fashion, Design and Technology	Berg Publishers
Lanehove L V [70]	Smart textiles for medicine and health care: Materials, Systems and Applications	Woodhead Publishing in Textiles
Song H-y et al. [71]	A study on the synthesis and electrical properties of cellulose acrylamidoxime copolymer	Polymer
Dias T [72]	Electronic Textiles: Smart Fabrics and Wearable Technology	Woodhead Publishing
Guo L et al. [73]	Systematic review of textile-based electrodes for long-term and continuous surface electromyography recording	Text. Res. J
Oindrila H et al. [74]	Metal nanowires grown in situ on polymeric fibres for electronic textiles	Nanoscale Adv
Wangcheng L et al. [75]	Wet-Spun Side-by-Side Electrically Conductive Composite Fibers	ACS Appl. Electron. Mater
Wendler J et al. [76]	Novel textile moisture sensors based on multi-layered braiding constructions	Text. Res. J.
Lah A Š et al. [77]	A NiTi alloy weft knitted fabric for smart firefighting clothing	Smart Mater. Struct
Chena S P et al. [78]	Highly stretchable and conductive silver nanowire thin films formed by soldering nanomesh junctions	Phys. Chem. Chem. Phys
Lu Y et al. [79]	Elastic, conductive, polymeric Hydrogelsand sponges	Sci. Rep
Standoli C E et al. [80]	A Smart Wearable sensor system for counter-fighting overweight in teenagers	Sensors (Basel)
Heo E et al. [81]	A wearable textile antenna for wireless power transfer by magnetic resonance	Text. Res. J.
Belmonte-Fernández Ó et al. [82]	An indoor positioning system based on wearables for ambient-assisted living	Sensors
Seo H I et al. [83]	reparation and characterization of Carbon aerogel activated with KOH and CO− effect of pore size distribution on electrochemical properties as EDLC electrodes-	Polymer
Yang C et al. [84]	Innovative artificial leather with high thermal conductivity as a new leather product	Text. Res. J
Yang C et al. [85]	A novel approach for developing high thermal conductive artificial leather by utilizing smart electronic materials	Text. Res. J
Bedeloglu A C et al. [86]	A photovoltaic fiber design for smart textiles	Text. Res. J.
Li L et al. [87]	A resistive network model for conductive knitting stitches	Text. Res. J.
Wang H et al. [88]	Bioinspired fluffy fabric with in situ grown carbon nanotubes for ultrasensitive wearable airflow sensor	Adv. Mater.
Liang X et al. [89]	Stable and biocompatible carbon nanotube ink mediated by silk protein for printed electronics	Adv. Mater
Fujiwara A et al. [90]	Local electronic transport through a junction of SWNT bundles	Phys. B
Lourie O et al. [91]	Evidence of stress transfer and formation of fracture clusters in carbon nanotube-based composites	Compos. Sci. Technol
Lim H-k et al. [92]	Dispersity of CNT and GNF on the Polyurethane matrix: Effect of Polyurethane chemical structure	Polymer
Oh Y et al. [93]	Investigation of Mechanical and Electrical Properties of Hybrid Composites Reinforced with Carbon Nanotubes and Micrometer-Sized Silica Particles	Korean Society of Mechanical Engineers
Park S H et al. [94]	Effect of Dispersion Control of Multi-walled Carbon nanotube in High Filler Content nanocomposite Paste for the Fabrication of counter Electrode in Dye-sensitized Solar Cell	Polymer
Ko J H et al. [95]	Ultrahigh molecular weight polyethylene hybrid films with functionalized-MWNT: Thermomechanical properties, morphology, gas permeability, and optical transparency	Polymer
Kim H et al. [96]	Preparation, morphology and electrical conductivity of polystyrene/polydopamine-carbon nanotube microcellular foams via high internal phase emulsion polymerization	Polymer
Jung J K et al. [97]	Electrical resistivity and mechanical properties of polypropylene composites containing carbon nanotubes and stainless steel short fibers	Polymer
Chen Y et al. [98]	Artificial synapses based on nanomaterials	Nanotechnology
Carrara S [99]	Nano-bio-technology and sensing chips: New systems for detection in personalized therapies and cell biology	Sensors (Basel)
Doopedia [100]	Carbon nanotube	Knowledge Encyclopedia of Naver
Apichit M et al. [101]	Influence of multi-walled carbon nanotubes reinforced honeycomb core on vibration and damping responses of carbon fiber composite sandwich shell structures	Polymer Composites
Shemshadi R et al. [102]	A smart thermoregulatory nanocomposite membrane with improved thermal properties: Simultaneous use of graphene family and micro-encapsulated phase change material	Text. Res. J

**Table 6 materials-16-01351-t006:** Systematization/enumeration of shape memory papers with bibliographic source.

Authors	Title	Bibliographic Source
Han H R et al. [103]	Shape memory and breathable waterproof properties of polyurethane nanowebs	Text. Res. J.
Meng Q et al. [104]	Biological evaluations of a smart shape memory fabric	Text. Res. J.
Mather P T et al. [105]	Shape memory polymer research	Annu. Rev. Mater. Res.
Mattila H R [106]	North America Intelligent Textiles and Clothing	CRC Press
Zhao Z et al. [107]	3D printing of a thermoplastic shape memory polymer using FDM	Meeting Conference Paper, in, APS March
Razzaq M Y et al. [108]	Thermomechanical studies of aluminum nitride filled shape memory polymer composites	Polym. Compos.
Matsumoto H et al. [109]	Shape-memory properties of electrospun nonwoven fabrics prepared from degradable polyesterurethanes containing poly(x-pentadecalactone) hard segments	Eur. Polym. J.
Michalak M et al. [110]	A smart fabric with increased insulating properties	Text. Res. J.
Michalak M et al. [111]	A smart textile fabric with two-way action	Text. Res. J.

## Data Availability

Not applicable

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
