# Peer review of "Hybrid Fiber Materials according to the Manufacturing Technology Methods and IOT Materials: A Systematic Review"

_materials, 2023, doi:10.3390/ma16041351_

Round 1

Reviewer 1 Report

1) International standards of wearable technology refer to earlier dates, so it is recommended to refer to newer standards.

2) The use of dashes in Table 1 is not standard, carefully check and modify.

3) In the aspect of 3.2 electrospinning, the application field of electrostatic spinning and the factors affecting electrostatic spinning are mainly expounded. The introduction of electrostatic spinning preparation of mixed fibers is relatively less, and should be supplemented.

4) It is doubtful that cellulose solution of NMMO is the best system for electrostatic stable formation of nanofibers.

5) Some important papers concerning intelligent wearable devices based on electrospinning hybrid nanofibers are worth of introduction to improve its readability, such as : Chem. Eng. J. 450 (2022) 137937. and Carbon Energy 4 (2022) 621-632.

Author Response

Response to Reviewer 1 Comments

Thank you for your kind comments for the paper. Details of the corrections are as follows. And I received 'MDPI english editing' throughout the thesis.

The attached file is largely divided into three parts. 1) Reviewer's Answer 2) Paper Version 1: The blue text in the text is a correction to the reviewer's answer. 3) Paper Version 2: The red text of the text shows both before and after modification of the text, including mdpi English editing. Thank you again for your review.

1) International standards of wearable technology refer to earlier dates, so it is recommended to refer to newer standards.

Response 1: I wish to express my gratitude to your comments. I revised the advice you gave me. The modified abstract are as follows:

Abstract: With the development of convergence technology, the Fourth Industrial Revolution, information technology (IT), the Internet of Things (IoT), and artificial intelligence (AI), there has been increasing interest in the materials industry. In recent years, numerous studies have attempted to identify and explore multi-functional cutting-edge hybrid materials. In this paper, the international literature on the materials used in hybrid fibers and manufacturing technologies are investigated and their future utilization in the industry is predicted. And, a systematic review is also conducted. This includes sputtering, electrospun nanofibers, 3D (three-dimensional) printing, shape memory, conductive materials. Sputtering technology is an eco-friendly, intelligent material that does not use water and can be applied as an advantageous military stealth material, conductive materials, electromagnetic blocking materials and etc. Electrospinning can be applied to breathable fabrics, filtration, protective clothing with high moisture vapor transport, toxic chamical re-sistance, fibrous drug delivery sys-tems, nanoliposomes and etc. 3D printing can be used in various fields, such as core-sheath fibers, artificial organs and etc. Conductive materials include metal nanowires, polypyrrole, polyaniline, and CNT(Carbon Nano Tube), and can be used in sensors, actuators, and light-emitting devices. When shape-memory materials deform into a temporary shape, they can return to their original shape in response to external stimuli. Shape-memory materials are, therefore, used in medical fields in certain applications, such as surgical sutures. This study attempted to examine in-depth hybrid fiber materials and manu-facturing technologies. And, This study systematically presents materials of various applications for future IOT hybrid fibers.

2) The use of dashes in Table 1 is not standard, carefully check and modify.

Response 2: Thank you for your astute comments. I revised the Table 1. The modified Table1 is as follows:

Table 1. Status of major international standards being established in TC124 related to wearable technology [45].

Standard number

(include status)

Standard name

Proposed year

IEC 63203-101-1:2021

Wearable electronic devices and technologies - Part 101-1: Terminology

2021

IEC 63203-204-1:2021

Wearable electronic devices and technologies - Part 204-1: Electronic textile - Test method for assessing washing durability of leisurewear and sportswear e-textile systems

2021

IEC 63203-401-1 ED1

Wearable electronic devices and technologies - Part 401-1: Devices and Systems – Functional elements - Evaluation method of the stretchable resistive strain sensor

2019

IEC 63203-402-1:2022

Wearable electronic devices and technologies - Part 402-1: Performance measurement of fitness wearables - Test methods of glove-type motion sensors for measuring finger movements

2022

BS EN IEC 63203-406-1:2022

Wearable electronic devices and technologies. Test method for measuring surface temperature of wrist-worn wearable electronic devices while in contact with human skin (IEC 63203-406-1:2021).

2022

IEC TR 63203-250-1:2021

Wearable electronic devices and technologies - Part 250-1: Electronic textile - Snap fastener connectors between e-textiles and detachable electronic devices

2021

3) In the aspect of 3.2 electrospinning, the application field of electrostatic spinning and the factors affecting electrostatic spinning are mainly expounded. The introduction of electrostatic spinning preparation of mixed fibers is relatively less, and should be supplemented.

Response 3: Thank you for your astute comments. I insert the sentences(with reference ‘[61], [62]’) in main text as follows. And I revised the all references.

Page 7 Line 263: In addition, recently, studies conducted by electrospinning using hybrid materials mixed with the two have been actively conducted.

Amjadi et al. studied carbone-loaded zein/pullulan hybrid electrospun nanofibers for food and medical use. Zein/pullulan hybrid nanoofiber was produced in three mix-ing ratios: 90:10, 80:20, and 70:30 by the electrospinning method. The tensile strength and water contact angle values of the zein/pullulan/carvone nanofiber were 7.09 ± 0.85 MPa and 96.6 ± 0.7° respectively. In addition, the incorporation of carbon into NF pro-vided DPPH elimination activity (9.6 ± 1.2%) and inhibitory activity against S. Aureus (39.95 ± 3.81 mm) and E. coli (12.2 ± 1.4 mm) bacteria. The biocompatibility of the de-veloped NP has been approved by in vitro cytotoxicity analysis. In conclusion, the de-veloped electrospinning NFs have significant potential in several applications, such as food packaging and wound dressing [61].

Chiu et al. used a double nozzle electrospinning process side by side to manufac-ture flexible hybrid electronics (FHE) materials with excellent elasticity. Very stable electrical conductivity was also provided to the membrane electrode produced by us-ing silver nanoparticles (AgNP) and carbon-based nanomaterials having a different structure. The AgNP/carbon nanomaterial was coated on a binary polymer nanofiber composed of polyurethane (PU) and polyvinylidene difluoride (PVDF) on a nanofiber membrane. FHE nanofiber electrodes were finally incorporated into garments designed to accurately measure human body detection signals (e.g., ECG and EMG) signals). AgNP/graphene oxide (GO) nanofiber electrodes showed a continuous phase with a stable material microstructure after 5000 iterations of a 50% tension-tension fatigue test. The waveform patterns obtained from the proposed AgNP/GO nanofiber elec-trodes were compared with conventional ECG and EMG electrodes. Nanofiber web electrodes treated with organic/inorganic mixed dispersants and verified through electrical and fatigue characteristics tests are suitable for long-term ECG and EMG monitoring and have excellent potential in wearable smart sensors [62].

4) It is doubtful that cellulose solution of NMMO is the best system for electrostatic stable formation of nanofibers.

Response 4: I insert the sentences as follows.

Page 7 line 246: Based on studies conducted with phosphoric acid mixtures not commonly used in the industry, Laszkiewicz et al. argued that the cellulose solution of NMMO(N-Methylmorpholine N-oxide) could be demonstrated to be an good system for the stable formation of nanofibers using static electricity [3].

5) Some important papers concerning intelligent wearable devices based on electrospinning hybrid nanofibers are worth of introduction to improve its readability, such as : Chem. Eng. J. 450 (2022) 137937. and Carbon Energy 4 (2022) 621-632.

Response 5: I insert the sentences(with reference ‘[59], [60]’) in main text as follows. And I revised the all references.

Page 7 Line 250: He et al. suggested thermoelectric nanofiber yarns using CNT/PEDOT: PSS with electrospinning technology, which showed high stretchability (~350) and seamability. In addition, self-powered strain sensors consisting of threads show corresponding thermal voltage changes according to strain, which can be used to optimize the shoot-ing rate of basketball players. These unique features have shown that thermoelectric nanofiber seals have a wide range of prospects in smart wearables such as wearable generators, respiratory monitoring, and motor optimization [59].

He et al. proposed a state-of-the-art manufacturing strategy that combines elec-trospinning technology and spraying technology to manufacture (2022) carbon nano-tube (CNT)/polyvinylpyrrolidone (PVP)/polyurethane (PU) composite thermoelectric fabric. As a result, a self-powered sensor for detecting joint motion has been success-fully proposed. In addition, the electrical conductivity and Sebeck coefficient did not change after 1000 bending times [60].

Thank you again for reviewing my thesis well.

Reviewer 2 Report

 I have carefully read this article and I believe that before publication it needs a major revision for the following reasons:

To improve the way of presentation, I recommend the authors to take into account the following aspects:

- the systematization/enumeration of all types of materials that they presented in the article should be highlighted first in the form of a table, indicating the authors or the bibliographic source, then discussed in detail;

- all abbreviations used in the article must be explained in parentheses; although most of them are explained, there are still some that cannot be understood by beginners or uninitiated readers in this field.

- citations in the text are made using the surname and not the first name of the author: examples: ELena for source [58]; MoniKa for source {3]. However, in this last case, there is no Monika name in source 3, so the authors used an erroneous citation.

- I checked the association of the authors' names with the bibliographic source number, and found inconsistencies for:

Weimin for [54]

Bogumi for [3]

Chen for [68, 94-97]

- for Li et. al, the bibliographic source was not indicated at the end of the sentence [83].

I recommend the authors to carefully check the names of the cited authors, their correct association with the related number in the References and respecting the chronology in the act of citation.

Author Response

Response to Reviewer 2 Comments

I have carefully read this article and I believe that before publication it needs a major revision for the following reasons:

To improve the way of presentation, I recommend the authors to take into account the following aspects:

Thank you for your kind comments for the paper. Details of the corrections are as follows. And I received 'MDPI english editing' throughout the thesis. The attached file is largely divided into three parts. 1) Reviewer's Answer 2) Paper Version 1: The blue text in the text is a correction to the reviewer's answer. 3) Paper Version 2: The red text of the text shows both before and after modification of the text, including mdpi English editing. Thank you again for your review.

1) the systematization/enumeration of all types of materials that they presented in the article should be highlighted first in the form of a table, indicating the authors or the bibliographic source, then discussed in detail;

Response: I wish to express my gratitude to your comments. Table 2~6 were inserted as recommended.

Table 2. Systematization/enumeration of sputtering papers with bibliographic source

Authors

Title

bibliographic source

[47] Su C-I et al

Performance of viscose rayon based activated carbon fabric modified by sputtering silver and continuous plasma treatment

Text. Res. J

[48] Depla D et al

Smart textiles: An explorative study of the use of magnetron sputter deposition

Text. Res. J

[49] He S  et al

Surface functionalization of Ag/polypyrrole-coated cotton fabric by in situ polymerization and magnetron sputtering

Text. Res. J

[50]  Yuan X et al

Electrical and optical properties of polyester fabric coated with Ag/TiO2 composite films by magnetron sputtering

Text. Res. J

[51] Han H R et al

A study on thermal and physical properties of nylon fabric treated by metal sputtering (Al, Cu, Ni)

Text. Res. J

[52] Han H R

Characteristics of infrared blocking, stealth and color difference of aluminum sputtered fabrics

J. Korean Soc. Cloth. Text

[53] Wu et al

A facile method to prepare a wearable pressure sensor based on fabric electrodes for human motion monitoring

Text. Res. J

Table 3. Systematization/enumeration of electrospinning papers with bibliographic source

Authors

Title

bibliographic source

[54] Kang W et al

2011 A new method for preparing alumina nanofibers by electrospinning technology

Text. Res. J.

[55]  Cavaliere S et al

Electrospinning for Advanced Energy and Environmental Applications

CRC Press

[56]  Ding B et al

Electrospun Nanofibers for Energy and Environmental Applications

Springer

[57] Dávila S M et al

Challenges and advantages of electrospun nanofibers in agriculture: A review

Mater. Res. Express

[58]  Bokova E S et al

Obtaining new biopolymer materials by electrospinning

Fibres Text. East. Eur

[59]  He X et al

Continuous manufacture of stretchable and integratable thermoelectric nanofiber yarn for human body energy harvesting and self-powered motion detection

Chemical Engineering Journal

[60]  He X et al

Highly stretchable, durable, and breathable thermoelectric fabrics for human body energy harvesting and sensing

Carbon energy

[61]  Amjadi S et al

Development and characterization of the carvone-loaded zein/pullulan hybrid electrospun nanofibers for food and medical applications

Industrial Crops and Products

[62]  Chiu CW et al

2022 Flexible Hybrid Electronics Nanofiber Electrodes with Excellent Stretchability and Highly Stable Electrical Conductivity for Smart Clothing

ACS Appl. Mater. Interfaces

Table 4. Systematization/enumeration of three-dimentional papers with bibliographic source

Authors

Title

bibliographic source

[63]  Zaharin H A et al

Additive Manufacturing Technology for Biomedical Components: A review

IOP Conf. Ser.: Mater. Sci. Eng.

[64]  Yi R et al

Delta DLP 3-D printing of large models

IEEE Trans. Autom. Sci. Eng.

[65]  Jung J-t et al

3D printer operators.

EBS broadcasting textbook Book Publishing Agency Gun Gi Won

[66]  Korea I et al

3D printer operations technician

Yeamoonsa.

[67] Encyclopedia

Selective laser sintering

Wikipedia the free encyclopedia

[68]  Giannopoulos A A et al

Applications of 3D printing in cardiovascular diseases

Nat. Rev. Cardiol

Table 5. Systematization/enumeration of conductive materials papers with bibliographic source

Authors

Title

bibliographic source

[69] Quinn B

Textile Futures: Fashion, Design and Technology

Berg Publishers

[70]  Lanehove L V

Smart textiles for medicine and health care: Materials, Systems and Applications

Woodhead Publishing in Textiles

[71]  Song H-y et al

A study on the synthesis and electrical properties of cellulose acrylamidoxime copolymer

Polymer

[72]  Dias T

Electrinic Textiles: Smart Fabrics and Wearale Technology

Woodhead Publishing

[73]  Guo L et al

Systematic review of textile-based electrodes for long-term and continuous surface electromyography recording

Text. Res. J

[74] Oindrila H et al

Metal nanowires grown in situ on polymeric fibres for electronic textiles

Nanoscale Adv

[75]  Wangcheng L et al

Wet-Spun Side-by-Side Electrically Conductive Composite Fibers

ACS Appl. Electron. Mater

[76] Wendler J et al

Novel textile moisture sensors based on multi-layered braiding constructions

Text. Res. J.

[77]  Lah A Š et al

A NiTi alloy weft knitted fabric for smart firefighting clothing

Smart Mater. Struct

[78]  Chena S P  et al

Highly stretchable and conductive silver nanowire thin films formed by soldering nanomesh junctions

Phys. Chem. Chem. Phys

[79]   Lu Y et al

Elastic, conductive, polymeric Hydrogelsand sponges

Sci. Rep

[80]   Standoli C E et al

A SmartWearable sensor system for counter-fighting overweight in teenagers

Sensors (Basel)

[81]  Heo E et al

A wearable textile antenna for wireless power transfer by magnetic resonance

Text. Res. J.

[82] Belmonte-Fernández Ó et al

An indoor positioning system based on wearables for ambient-assisted living

Sensors

[83]  Seo H I et al

reparation and characterization of Carbon aerogel activated with KOH and CO− effect of pore size distribution on electrochemical properties as EDLC electrodes-

Polymer

[84] Yang C  et al

Innovative artificial leather with high thermal conductivity as a new leather product

Text. Res. J

[85]  Yang C et al

A novel approach for developing high thermal conductive artificial leather by utilizing smart electronic materials

Text. Res. J

[86]  Bedeloglu A C et al

A photovoltaic fiber design for smart textiles

Text. Res. J.

[87]  Li L et al

A resistive network model for conductive knitting stitches

Text. Res. J.

[88]  Wang H et al

Bioinspired fluffy fabric with in situ grown carbon nanotubes for ultrasensitive wearable airflow sensor

Adv. Mater.

[89]  Liang X et al

Stable and biocompatible carbon nanotube ink mediated by silk protein for printed electronics

Adv. Mater

[90]  Fujiwara A et al

Local electronic transport through a junction of SWNT bundles

Phys. B

[91]  Lourie O et al

Evidence of stress transfer and formation of fracture clusters in carbon nanotube-based composites

Compos. Sci. Technol

[92]  Lim H-k et al

Dispersity of CNT and GNF on the Polyurethane matrix: Effect of Polyurethane chemical structure

Polymer

[93]  Oh Y et al

Investigation of Mechanical and Electrical Properties of Hybrid Composites Reinforced with Carbon Nanotubes and Micrometer-Sized Silica Particles

Korean Society of Mechanical Engineers

[94]  Park S H et al

Effect of Dispersion Control of Multi-walled Carbon nanotube in High Filler Content nanocomposite Paste for the Fabrication of counter Electrode in Dye-sensitized Solar Cell

Polymer

[95]  Ko J H et al

Ultrahigh molecular weight polyethylene hybrid films with functionalized-MWNT: Thermomechanical properties, morphology, gas permeability, and optical transparency

Polymer

[96]  Kim H et al

Preparation, morphology and electrical conductivity of polystyrene/polydopamine-carbon nanotube microcellular foams via high internal phase emulsion polymerization

Polymer

[97]  Jung J K et al

Electrical resistivity and mechanical properties of polypropylene composites containing carbon nanotubes and stainless steel short fibers

Polymer

[98]  Chen Y et al

Artificial synapses based on nanomaterials

Nanotechnology

[99]  Carrara S

Nano-bio-technology and sensing chips: New systems for detection in personalized therapies and cell biology

Sensors (Basel)

[100] Doopedia

Carbon nanotube

Knowledge Encyclopedia of Naver

[101]  Apichit M et al

Influence of multi-walled carbon nanotubes reinforced honeycomb core on vibration and damping responses of carbon fiber composite sandwich shell structures

Polymer Composites

[102]  Shemshadi R et al

A smart thermoregulatory nanocomposite membrane with improved thermal properties: Simultaneous use of graphene family and micro-encapsulated phase change material

Text. Res. J

Table 6. Systematization/enumeration of Shape memory papers with bibliographic source

Authors

Title

bibliographic source

[103] Han H R et al

Shape memory and breathable waterproof properties of polyurethane nanowebs

Text. Res. J.

[104]  Meng Q et al

Biological evaluations of a smart shape memory fabric

Text. Res. J.

[105] Mather P T et al

Shape memory polymer research

Annu. Rev. Mater. Res.

[106]  Mattila H R

North america Intelligent Textiles and Clothing

CRC Press

[107]  Zhao Z et al

3D printing of a thermoplastic shape memory polymer using FDM

Meeting Conference Paper, in, APS March

[108]  Razzaq M Y et al

Thermomechanical studies of aluminum nitride filled shape memory polymer composites

Polym. Compos.

[109]  Matsumoto H et al

Shape-memory properties of electrospun nonwoven fabrics prepared from degradable polyesterurethanes containing poly(x-pentadecalactone) hard segments

Eur. Polym. J.

[110]  Michalak M et al

A smart fabric with increased insulating properties

Text. Res. J.

[111]  Michalak M et al

A smart textile fabric with two-way action

Text. Res. J.

2) all abbreviations used in the article must be explained in parentheses; although most of them are explained, there are still some that cannot be understood by beginners or uninitiated readers in this field.

Response: I wish to express my gratitude to your comments. All abbreviations in the text are written in full names.

3) citations in the text are made using the surname and not the first name of the author: examples: ELena for source [58]; MoniKa for source {3]. However, in this last case, there is no Monika name in source 3, so the authors used an erroneous citation.

Response: It was adjusted as recommended.

* Page7 Line230: Bokova et al. addressed fiber electrical rotation technology for nonwoven fabric pro-duction in various applications. In particular, they studied the conditions for forming nano- and microfibers in collagen hydrolysate and dibutyrylchitine solutions, as well as polymer complexes based on polyacrylic acid, polyvinyl alcohol, and polyethylene oxide. Comparative analyses of electrical rotations, electrical capillary tubes, and elec-trical nanospiders were performed. The results show promise not only for garment and shoe production, but also for the application of nonwoven fabrics in pharmaceutical hygiene practices [58].

* Deleted for 'Monika' in the text.

4) I checked the association of the authors' names with the bibliographic source number, and found inconsistencies for:

Weimin for [54]

Response: It was adjusted as recommended.

Page 7 line 237: Kang et al

Bogumi for [3]

Response: It was adjusted as recommended.

Page 7 line 242: Laszkiewicz et al

Chen for [68, 94-97]

Response: It was adjusted as follows.

Page 15 line 523: discrete artificial synapses. LTP(long-term potentiation) or LTD(long-term depression) was induced by the increased conductivity as a result of the dehydrogenation of CNTs. LTP or LTD was not obtained when the gap between two consecutive spikes was wider than 20 ms. This was because the hydrogen ions in PEG(poly(ethylene glycol)) returned to the equilibrium position within that interval. The CNT synaptic device consumed only 7.5p J/spike, which is compatible with the widespread integration of artificial neural networks [98].

5) for Li et. al, the bibliographic source was not indicated at the end of the sentence [83].

Response: It was inserted as follows.

Page 15 line 498: [87]

6) I recommend the authors to carefully check the names of the cited authors, their correct association with the related number in the References and respecting the chronology in the act of citation.

Response: All references were checked as recommended.

Thank you again for reviewing my thesis well.

Reviewer 3 Report

The current manuscript presents a review of hybrid fiber materials and their manufacturing technologies. There are some issues to be considered as follows:

- The abstract should be revised to concisely present the impact and main target of the current study and the proposed material including their applications and characteristics. 

- The expression of "hybrid fiber materials" should be defined in more detail, in addition to obviously illustrating the categories of these materials and their direct applications.

- It is recommended to remove the subsections numbering in the introduction section. This section should include an introductory review of materials and methods.

- The problem statement should be well defined by the end of the introduction section. 

- It is recommended to remove section 2 and merge their data to the introduction section. 

- An applicable example should be included for each manufacturing technique in section 3.

- At lines 200 and 205, there is no need to redefine the abbreviations of FDM and DLP.

- In section 4, the expressions of "IOT hybrid fiber materials" should be well defined. 

- At line 355, the study of Li et. al should be cited, there is no reference stated for this paragraph. The same for the study stated in line 384.

- There is a significant lack of statistics and discussion of the hybrid fiber materials and their manufacturing methodologies.  

- The language and grammar should be carefully revised. Please try to avoid redundancy and long sentences.

- The conclusion section should include the current status, the main obstacles to be applied, and the future perspective. It is recommended to use a bullet points style to summarize proposed titles.   

Author Response

Response to Reviewer 3 Comments

The current manuscript presents a review of hybrid fiber materials and their manufacturing technologies. There are some issues to be considered as follows:

Thank you for your kind comments for the paper. Details of the corrections are as follows. And I received 'MDPI english editing' throughout the thesis. The attached file is largely divided into three parts. 1) Reviewer's Answer 2) Paper Version 1: The blue text in the text is a correction to the reviewer's answer. 3) Paper Version 2: The red text of the text shows both before and after modification of the text, including mdpi English editing. Thank you again for your review.

1) The abstract should be revised to concisely present the impact and main target of the current study and the proposed material including their applications and characteristics.

Response: It was adjusted as recommended.

Abstract: With the development of convergence technology, the Fourth Industrial Revolution, in-formation technology (IT), the Internet of Things (IoT), and artificial intelligence (AI), there has been increasing interest in the materials industry. In recent years, numerous studies have attempted to identify and explore multi-functional cutting-edge hybrid materials. In this paper, the international literature on the materials used in hybrid fibers and manufacturing technologies are investigated and their future utilization in the industry is predicted. And, a systematic review is also conducted. This includes sputtering, electrospun nanofibers, 3D (three-dimensional) printing, shape memory, conductive materials. Sputtering technology is an eco-friendly, intelligent material that does not use water and can be applied as an advantageous military stealth material, conductive materials, electromagnetic blocking materials and etc. Electrospinning can be applied to breathable fabrics, filtration, protective clothing with high moisture vapor transport, toxic chamical re-sistance, fibrous drug delivery systems, nanoliposomes and etc. 3D printing can be used in various fields, such as core-sheath fibers, artificial organs and etc. Conductive materials include metal nanowires, polypyrrole, polyaniline, and CNT(Carbon Nano Tube), and can be used in sensors, actuators, and light-emitting devices. When shape-memory materials deform into a temporary shape, they can return to their original shape in response to external stimuli. Shape-memory materials are, therefore, used in medical fields in certain applications, such as surgical sutures. This study attempted to examine in-depth hybrid fiber materials and manu-facturing technologies. And, This study sys-tematically presents materials of various applications for future IOT hybrid fibers.

2) The expression of "hybrid fiber materials" should be defined in more detail, in addition to obviously illustrating the categories of these materials and their direct applications.

Response: It was inserteded as recommended. Tables 2 to 6 are also added.

Page 9 Line 356: 3. IOT Hybrid Fiber Materials

With the advent of artificial intelligence and IT technology, this paper will exam-ine materials that can be used in IOT hybrid fibers that respond to external stimuli or have electrical conductivity.

3) It is recommended to remove the subsections numbering in the introduction section. This section should include an introductory review of materials and methods.

Response: It was adjusted as recommended.

4) The problem statement should be well defined by the end of the introduction section.

Response: It was inserteded as follows.

Page 4 line 130: Recently, demand for multifunctional materials is increasing due to the advent of high-tech and the advancement of consumer needs. However, there are very few data that have been systematically classified and organized by collecting extensive data on hybrid fiber materials and production technologies. Therefore, this study attempted to examine in-depth hybrid fiber materials and manufacturing technologies.

5) It is recommended to remove section 2 and merge their data to the introduction section.

Response: It was adjusted as recommended. All section numbers were changed.

6) An applicable example should be included for each manufacturing technique in section 3.

Response: It was inserted as recommended.

* Page 6 line 168: Applicable examples of using sputtering technology include semiconductors, au-tomotive parts, heavy industries, stealth clothing materials, electromagnetic wave blocking materials, electrically conductive materials, and sensors.

* Page 8 line 295: Applicable examples of using electrospinning technology include filtration, pro-tective clothing with high moisture vapor transport, toxic chamical resistance, breathability fabric, tissue engineering applications, wound dressings, fibrous drug de-livery systems, implants, transdermal patches, nanoliposomes and etc.

* Page 9 line 350: Applicable examples of using three-dimensional printing technology include ar-tificial organs, skulls, fetal figures, electronic parts, cars, buildings, shoes, accessories, dresses and etc.

7) At lines 200 and 205, there is no need to redefine the abbreviations of FDM and DLP.

Response: : It was adjusted as recommended.

8) In section 4, the expressions of "IOT hybrid fiber materials" should be well defined.

Response: Please provide your response for Point 2. (in red)

Page 9 Line 356: 3. IOT Hybrid Fiber Materials

With the advent of artificial intelligence and IT technology, this paper will exam-ine materials that can be used in IOT hybrid fibers that respond to external stimuli or have electrical conductivity.

9) At line 355, the study of Li et. al should be cited, there is no reference stated for this paragraph. The same for the study stated in line 384.

Response: It was adjusted as recommended.

10) There is a significant lack of statistics and discussion of the hybrid fiber materials and their manufacturing methodologies. 

Response: It was adjusted as recommended. It was added to the conclusion as well as the middle text(* Page 6 line 168, * Page 8 line 295,* Page 9 line 350)

* Page 6 line 168: Applicable examples of using sputtering technology include semiconductors, au-tomotive parts, heavy industries, stealth clothing materials, electromagnetic wave blocking materials, electrically conductive materials, and sensors.

* Page 8 line 295: Applicable examples of using electrospinning technology include filtration, pro-tective clothing with high moisture vapor transport, toxic chamical resistance, breathability fabric, tissue engineering applications, wound dressings, fibrous drug de-livery systems, implants, transdermal patches, nanoliposomes and etc.

* Page 9 line 350: Applicable examples of using three-dimensional printing technology include ar-tificial organs, skulls, fetal figures, electronic parts, cars, buildings, shoes, accessories, dresses and etc.

* Page 18 line 616:

               Current status

Sputtering technology is an eco-friendly, intelligent material that does not use water. In the case of sputtering, it can be applied as an EM shield, conductive fabric, and stealth material. Electrospinning technology using high voltage can be applied to breathable nanowebs, artificial blood vessels, etc. by manufacturing hybrid nanofibers. Three-dimensional printing includes FDM, DLP, SLS, and LOM methods. It is projected that 3D printing will be used in various fields, such as core-sheath fibers.

Conductive materials can be used in sensors, EM blocking, wearable computer and etc. And well known conductive materials include silver nanowires (AgNWs), polypyrrole (Ppy), PANI (polyaniline), carbon nanotubes (CNTs) and etc. When the shape-memory material deforms into a temporary shape, it can return to its original shape by changes, such as pH and temperature. Shape-memory materials can be used for intelligent textiles, surgical sutures and etc.

               The main obstacles to be applied

 In the case of electrospinning technology, 'limitations of mass production, dura-bility, and manufacturing speed ' have been raised as obstacle.

And in the case of 3D printing, the problem of toxicity is being raised during the manufacturing process.

In the case of electrically conductive materials, when electricity is applied to fab-rics, they should be worn only for a short period of time or with a small area in consid-eration of the harmfulness of electromagnetic waves directly touching the human body. Electrically conductive smart wear materials could have possible side effects on the human body. Additionally, further research should be conducted on the adequate use of space and time, and the prevention of electrocution during lightning storms

In the case of shape memory materials, durability, etc. should be considered when repeatedly used.

               Future perspective

As the demand for multifunctional materials increases, it is expected that more multifunctional hybrid fibers will emerge in the future, which are more hu-man-friendly, faster in production, and have good strength and washing durability. In particular, conductive fabric that combines technology and arts, fiber with improved medical function, hybrid fiber with increased human affinity, drug delivery system fi-ber with increased absorption rate, breathable fabric, artificial blood vessels, artificial organs, electrical conductivity fiber with more sensor sensitivity and biodegradable SMP hybrid fibers are expected to increase.

11) The language and grammar should be carefully revised. Please try to avoid redundancy and long sentences.

Response: All texts were revised as recommended. I received 'MDPI english editing' throughout the thesis.

12) The conclusion section should include the current status, the main obstacles to be applied, and the future perspective. It is recommended to use a bullet points style to summarize proposed titles.

Response: It was adjusted as recommended.

  1. Conclusions

In this study, the status of future hybrid fiber materials and manufacturing tech-nologies were analyzed, according to the material domain, and the application in the manufacturing of hybrid fiber products. Specially systematic review is also conducted. This includes sputtering, electrospun nanofibers, 3D (three-dimensional) printing, shape memory, conductive materials. Hybrid fiber material applications, such as sen-sors, vascular, e-textiles, devices, valves, and intelligent textiles were investigated. The characteristics and requirements of these high-performance materials could pave the way for future hybrid fiber materials.

               Current status

Sputtering technology is an eco-friendly, intelligent material that does not use water. In the case of sputtering, it can be applied as an EM shield, conductive fabric, and stealth material. Electrospinning technology using high voltage can be applied to breathable nanowebs, artificial blood vessels, etc. by manufacturing hybrid nanofibers. Three-dimensional printing includes FDM, DLP, SLS, and LOM methods. It is projected that 3D printing will be used in various fields, such as core-sheath fibers.

Conductive materials can be used in sensors, EM blocking, wearable computer and etc. And well known conductive materials include silver nanowires (AgNWs), polypyrrole (Ppy), PANI (polyaniline), carbon nanotubes (CNTs) and etc. When the shape-memory material deforms into a temporary shape, it can return to its original shape by changes, such as pH and temperature. Shape-memory materials can be used for intelligent textiles, surgical sutures and etc.

               The main obstacles to be applied

 In the case of electrospinning technology, 'limitations of mass production, dura-bility, and manufacturing speed ' have been raised as obstacle.

And in the case of 3D printing, the problem of toxicity is being raised during the manufacturing process.

In the case of electrically conductive materials, when electricity is applied to fab-rics, they should be worn only for a short period of time or with a small area in consid-eration of the harmfulness of electromagnetic waves directly touching the human body. Electrically conductive smart wear materials could have possible side effects on the human body. Additionally, further research should be conducted on the adequate use of space and time, and the prevention of electrocution during lightning storms

In the case of shape memory materials, durability, etc. should be considered when repeatedly used.

               Future perspective

As the demand for multifunctional materials increases, it is expected that more multifunctional hybrid fibers will emerge in the future, which are more hu-man-friendly, faster in production, and have good strength and washing durability. In particular, conductive fabric that combines technology and arts, fiber with improved medical function, hybrid fiber with increased human affinity, drug delivery system fi-ber with increased absorption rate, breathable fabric, artificial blood vessels, artificial organs, electrical conductivity fiber with more sensor sensitivity and biodegradable SMP hybrid fibers are expected to increase.

Hybrid fiber materials are expected to significantly contribute to several areas of research. In this study, a foundation for artificial intelligence was built by analyzing the characteristics of hybrid fibers and their manufacturing technology methods, re-cent applications, demand, and related high-performance materials.

Thank you again for reviewing my thesis well.

Round 2

Reviewer 2 Report

In this form the article can be published.

Author Response

Cover letter

Thank you for your kind comments for the paper. Details of the corrections are as follows. And I received 'MDPI english editing' throughout the thesis. The attached file is largely divided into three parts. 1) Answer to the Reviewer’s comments 2) Paper Version 1: The green text in the paper is a correction to the answer to the second(2022.01.31) reviewer’s comment. The blue text in the paper is a correction to the answer to the first(2022.01.19) reviewer’s comment. 3) paper version 2: The red text of the text shows both before and after modification of the text, including mdpi English editing. Thank you again for your review.

Response to Reviewer 2 Comments

In this form the article can be published.

Response 1: I wish to express my gratitude to your opinion. I looked at English one more time as a whole thesis. Thank you again for your comments. Best wishes.

Reviewer 3 Report

Thanks to the author efforts, the revised manuscript is significantly improved. The review comments and recommendations are well addressed. However, there are some minor issues to be considered as follows:

- The abstract is still relatively long, it should not exceed 200 word. Please try try make the abstract focused and concise. 

- It is recommended to change the title of conclusion section to ' Summary and future prospective'.

- The conclusion section needs to be well revised and organized.  

Author Response

Cover letter

Thank you for your kind comments for the paper. Details of the corrections are as follows. And I received 'MDPI english editing' throughout the thesis. The attached file is largely divided into three parts. 1) Answer to the Reviewer’s comments 2) Paper Version 1: The green text in the paper is a correction to the answer to the second(2022.01.31) reviewer’s comment. The blue text in the paper is a correction to the answer to the first(2022.01.19) reviewer’s comment. 3) paper version 2: The red text of the text shows both before and after modification of the text, including mdpi English editing. Thank you again for your review.

Response to Reviewer 3 Comments

Thanks to the author efforts, the revised manuscript is significantly improved. The review comments and recommendations are well addressed. However, there are some minor issues to be considered as follows:

1) The abstract is still relatively long, it should not exceed 200 word. Please try try make the abstract focused and concise.

Response 1: I wish to express my gratitude to your opinion. . I revised the manuscript. The modified abstract is as follows:

Abstract: With the development of convergence technology, the Internet of Things (IoT), and arti-ficial intelligence (AI), there has been increasing interest in the materials industry. In recent years, numerous studies have attempted to identify and explore multi-functional cutting-edge hybrid materials. In this paper, the international literature on the materials used in hybrid fibers and manufacturing technologies were investigated and their future utilization in the industry is pre-dicted. And, a systematic review is also conducted. This includes sputtering, electrospun nanofibers, 3D (three-dimensional) printing, shape memory, and conductive materials. Sputtering technology is an eco-friendly, intelligent material that does not use water and can be applied as an advantageous military stealth material, electromagnetic blocking material and etc. Electrospinning can be applied to breathable fabrics, toxic chemical resistance, fibrous drug delivery systems, nanoliposomes and etc. 3D printing can be used in various fields, such as core-sheath fibers, artificial organs and etc. Conductive materials include metal nanowires, polypyrrole, polyaniline, and CNT(Carbon Nano Tube), and can be used in actuators, and light-emitting devices. When shape-memory materials deform into a temporary shape, they can return to their original shape in response to external stimuli. This study attempted to examine in-depth hybrid fiber materials and manufacturing technologies.

2)  It is recommended to change the title of conclusion section to ' Summary and future prospective'.

Response 2: I wish to express my gratitude to your opinion. . I revised the manuscript. It was adjusted as follows.

Page 18 line 602: 4. Summary and future prospective

3) The conclusion section needs to be well revised and organized. 

Response 3: I wish to express my gratitude to your comments. I revised the manuscript. It was adjusted as follows.

Page 18 line 602: 4. Summary and future prospective

In this study, the status of hybrid fiber materials and manufacturing technologies were analyzed, according to the material domain, and the application in the manu-facturing of hybrid fiber products. An Especially systematic review is also conducted. This includes sputtering, electrospun nanofibers, 3D (three-dimensional) printing, shape memory, and conductive materials. This study has shown that hybrid fiber ma-terial applications, such as sensors, vascular, e-textiles, devices, valves, and intelligent textiles were investigated. The characteristics and requirements of these high-performance materials could pave the way for future hybrid fiber materials.

               Current status

Sputtering technology is an eco-friendly, intelligent material that does not use water. In the case of sputtering, it can be applied as an EM shield, conductive fabric, and stealth material. Electrospinning technology using high voltage can be applied to breathable nanowebs, artificial blood vessels, etc. by manufacturing hybrid nanofibers. Three-dimensional printing includes FDM, DLP, SLS, and LOM methods. It is projected that 3D printing will be used in various fields, such as core-sheath fibers.

Conductive materials can be used in sensors, EM blocking, wearable computers and etc. And well known conductive materials include silver nanowires (AgNWs), polypyrrole (Ppy), PANI (polyaniline), carbon nanotubes (CNTs) and etc. When the shape-memory material deforms into a temporary shape, it can return to its original shape by changes, such as pH, temperature and etc. Shape-memory materials can be used for intelligent textiles, surgical sutures and etc.

               The main obstacles to be applied

 In the case of electrospinning technology, 'limitations of mass production, dura-bility, and manufacturing speed ' have been raised as an obstacle.

And in the case of 3D printing, the problem of toxicity is raised during the manu-facturing process.

In the case of electrically conductive materials, when electricity is applied to fab-rics, they should be worn only for a short period or with a small area in consideration of the harmfulness of electromagnetic waves directly touching the human body. Elec-trically conductive smart wear materials could have possible side effects on the human body. Additionally, further research should be conducted on the adequate use of space and time, and the prevention of electrocution during lightning storms

In the case of shape memory materials, durability, etc. should be considered when repeatedly used.

               Future perspective

As the demand for multifunctional materials increases, it is expected that more multifunctional hybrid fibers will emerge in the future, which are more hu-man-friendly, faster in production, and have good strength and washing durability. In particular, conductive fabric that combines technology and arts, fiber with improved medical function, hybrid fiber with increased human affinity, drug delivery system fi-ber with increased absorption rate, breathable fabric, artificial blood vessels, artificial organs, electrical conductivity fiber with more sensor sensitivity and biodegradable SMP hybrid fibers are expected to increase.

Having made these brief points, it is time to bring this paper to a close and to end. Hybrid fiber materials are expected to significantly contribute to several areas of re-search. In this study, a foundation for artificial intelligence was suggested by analyzing the characteristics of hybrid fibers and their manufacturing technology methods, re-cent applications, demand, and related high-performance materials.

Thank you again for your comments. Best wishes.
